# A Thermal Tactile Sensation Display with Controllable Thermal Conductivity

**DOI:** 10.3390/mi10060359

**Published:** 2019-05-29

**Authors:** Seiya Hirai, Norihisa Miki

**Affiliations:** Department of Mechanical Engineering, Keio University, 3-14-1 Hiyoshi, Kohoku-ku, Yokohama, Kanagawa 223-8522, Japan; seiya_hirai@keio.jp

**Keywords:** tactile display, thermal tactile display, thermal sensation, thermal conductivity, liquid metal

## Abstract

We demonstrate a thermal tactile sensation display that can present various thermal sensations, namely cold/cool/warm/hot feelings, by varying the effective thermal conductivity of the display. Thermal sensation is one of the major tactile sensations and needs to be further investigated for advanced virtual reality/augmented reality (VR/AR) systems. Conventional thermal sensation displays present hot/cold sensations by changing the temperature of the display surface, whereas the proposed display is the first one that controls its effective thermal conductivity. The device contains an air cavity and liquid metal that have low and high thermal conductivity, respectively. When the liquid metal is introduced to fill up the air cavity, the apparent thermal conductivity of the device increases. This difference in the thermal conductivity leads to the users experiencing different thermal tactile sensations. Using this device, the threshold to discriminate the effective thermal conductivity was experimentally deduced for the first time. This thermal tactile display can be a good platform for further study of thermal tactile sensation.

## 1. Introduction

Tactile displays have been studied to present pseudo-tactile sensations to users for advanced information communication technologies, such as efficient teleoperation and virtual reality/augmented reality (VR/AR) [1,2,3,4,5,6,7]. Tactile sensations are categorized into five sensations: wetness, roughness, hardness, pain sensation, and thermal sensation [8,9]. Roughness can be represented by the surface geometry. An array of micro-actuators can form various surface geometries, and therefore, many tactile devices to present various roughness sensation have been proposed, where microelectromechanical systems (MEMS) technologies have played an important role [10,11,12]. Hardness in tactile research is stiffness, to be precise. Stiffness is a material property and is difficult to control. In previous work, a magnetorheological fluid was encapsulated inside flexible membranes, whose apparent stiffness could be varied with the external magnetic field [13,14,15]. The encapsulation process was developed, and the device was used as the stiffness distribution display. 

Thermal sensation is the sensation of cold/cool/warm/hot nature when we touch the surface of objects and plays a crucial role in tactile perception [16,17]. Conventionally, the thermal sensation displays present hot/cold surfaces by varying the surface temperature using Peltier elements [18,19,20]. The thermal module using the Peltier elements can be combined with other types of tactile displays, such as mechano-tactile display with an array of actuators [21,22] and electrostatic tactile display [23]. Thermal sensation display using radiation was proposed, which can stimulate subjects who are not in direct contact with the display [24]. Note that the thermal sensation is not determined only by the surface temperature, but also the thermal conductivity of the objects in contact [25]. For example, when we touch objects made of wood and metal which are at the same temperature, we perceive the metal to be colder than the wood, as illustrated in Figure 1. Materials with high thermal conductivity, such as metal, absorb heat from our finger, which leads to a cold/cool sensation. Control of thermal conductivity is challenging since it is an inherent physical property to each material and is determined by molecular configuration. Therefore, there have been no effective thermal tactile displays reported to date that can control their thermal conductivity.

In this work, we demonstrate a thermal tactile sensation display that can vary its thermal conductivity in a wide range and in an analogue manner by controlling its effective thermal conductivity. Effective thermal conductivity is the total thermal conductivity of the material and device, which depends not only on the material property, but also on the heterogeneous geometry. Since the temperatures of the device surface and the finger are room temperature (~25 °C) and body temperature (~37 °C), respectively, and are both low, we considered that the heat conduction through the device surface is dominant in the heat transfer. The conceptual sketch of the proposed tactile display is shown in Figure 2. The participant touches the display surface, which is a titanium plate. The display has an air cavity beneath the plate. Since the thermal conductivity of the air is as low as 0.024 W/mK at 0 °C, the effective thermal conductivity is also low at the original state. The display contains liquid metal encapsulated inside a flexible membrane. Liquid metal has equally high thermal conductivity to solid metals. As the liquid metal is supplied, it occupies the air cavity and the contact area between the liquid metal and the top plate increases. This increases the effective thermal conductivity of the display. The thermal property of this device was experimentally characterized and then perception tests were conducted to verify the effectiveness of this thermal tactile display. The property of thermal tactile perception was successfully characterized in a quantitative manner using the display. All the experimental protocols were approved by the Research Ethics Committee of Faculty of Science and Technology, Keio University.

## 2. Principle and Fabrication Process

### 2.1. Principle

The working principle and the structure of the device are shown in Figure 2 and Figure 3, respectively. The device encapsulates liquid metal with high thermal conductivity inside copper structures. The liquid metal we used for this device is Galinstan (68.5% gallium, 21.5% indium, and 10% tin), which has the thermal conductivity of 82 W/mK. It is sealed in the device with a latex rubber membrane with thermal conductivity of 0.13 W/mK. By controlling the amount of the liquid metal contacting the device’s surface, different thermal conductivities can be presented to a fingertip which is to be placed on the top surface. As shown in Figure 2, at the initial state, liquid metal is separated from the titanium surface, resulting in a low thermal conductivity, which is determined by the geometry of the copper structures. When the liquid metal is injected into the device from the syringe underneath, it expands spherically with the latex rubber and reaches the titanium surface. Here, the thermal conductivity of the latex rubber is assumed to be negligible since its thickness is as thin as 20 µm. The larger the contact area it becomes, the better the effective thermal conductivity the device possesses. The effective thermal conductivity can be increased until the whole surface is fully in contact with the liquid metal. The contact area and thus the effective thermal conductivity can be controlled in the range continuously.

### 2.2. Fabrication Process

Figure 4 shows the fabrication process of the thermal sensation tactile display. (a) Latex rubber is spin-coated onto a glass substrate at 1500 rpm for 30 s, which results in a membrane of 20 μm in thickness. Subsequently, the latex rubber was baked at 100 °C for 72 h. (b) An acrylic plate (20 mm × 20 mm × 1 mm) and a copper plate (20 mm × 20 mm × 5 mm) are processed with a numerical control (NC) cutting machine (MM-100, Modia Systems Co., Saitama Japan). Holes of 7 mm in diameter are drilled in a 2 × 2 array with intervals of 1 mm in the acrylic plate. In addition, two grooves with the size of 23 mm × 1 mm × 0.2 mm are formed on the acrylic plate, which pass through the center of the circles and work as the air escape paths when the air cavity decreases. Cavities of 17 mm × 17 mm × 2 mm and holes with diameters of 4.1 mm and 3 mm are processed onto a copper plate. (c) Both the acrylic plate and the copper plate are bonded to the latex rubber. (d) Liquid metal, Galinstan (Cool laboratory, Magdeburg, Germany), is injected into the cavity of the copper plate with a syringe. The syringe is connected to a luer fitting and a silicon tube so that the device can be set in a device holder. The cavity is sealed with polydimethylsiloxane (PDMS; Silpot 184 W/C, Dow Corning Toray Co., Ltd., Tokyo, Japan). (e) Finally, the titanium cover (20 mm × 20 mm × 0.05 mm) is bonded onto the acrylic plate of the device. 

## 3. Experimental Procedure

### 3.1. Measurement of the Contact Area

The relationship between the amount of injected liquid metal and the contact area of the liquid metal and the titanic plate was investigated with a microscope. In this experiment, to visualize the contact area, the titanium cover was replaced by a glass plate. 

### 3.2. Measurement of Effective Thermal Conductivity

The effective thermal conductivity of the device was measured with respect to the contact area using the flat plate comparison method (Figure 5) [26]. The effective thermal conductivity of the device was calculated using Fourier’s law, as described in Equation (1).
(1)$$Q=-A\lambda_{1}\frac{\left(T_{1}-T_{2}\right)}{l_{1}}=-A\lambda_{2}\frac{\left(T_{2}-T_{3}\right)}{l_{2}}$$
where Q is the heat transferred via the metal plate and the device. Since these materials are thermally insulated, the heat transfer quantity is constant. A is the contact area of the two materials. $\lambda_{1}$ is the thermal conductivity of a metal plate ($\lambda_{1}=83.5\;W/\mathrm{mK}$), and $\lambda_{2}$ is the thermal conductivity of the device. $T_{1}$, $T_{2}$, and $T_{3}$ are the measured temperatures at the interface between the metal plate and the hot plate, at the top surface of the device, and at the bottom of the device, respectively. $l_{1}$ and $l_{2}$ are the thicknesses of the metal plate ($l_{1}=5\;\mathrm{mm})$ and the device ($l_{2}=6\;\mathrm{mm})$, respectively. Note that all the other experiments were conducted at room temperature without the hot plate and the cool air.

### 3.3. Evaluation of Thermal Sensation

A sensory experiment was carried out with ten participants (21 to 24 years old, 8 males and 2 females) to investigate how the thermal conductivity of the device affects the thermal sensation. First, the participants were requested to touch the device when the injection amount of the liquid metal was 0.00 mL and 0.08 mL, i.e., when the hottest and coldest sensation were supposed to be presented. Then, the participants were requested to touch the device with the injection amount ranging from 0.00 mL to 0.08 mL and score the coldness on a seven-item scale from 1 (cold) to 7 (hot). 

### 3.4. Perceptual Threshold of Thermal Conductivity

Perception tests were conducted to deduce the perceptual threshold of thermal conductivity, if any, using the two-point identification method [27]. In this experiment, the injection amount was accurately controlled with a micro-syringe pump. We prepared two devices with different injection amount and thermal conductivity. The thermal conductivity of one device was fixed to be 75 W/mK, and the thermal conductivity of the other device was controlled between 80 W/mK and 105 W/mK with a step size of 5 W/mK in a random sequence. The participants were asked to touch the center of the device with the index fingers of both hands simultaneously, and then were asked whether they recognized a difference in coldness. To deduce the threshold more accurately, we conducted the same experiments varying the thermal conductivity with a step of 1 W/mK near the threshold. We consider that the initial temperatures of the finger and the device surface need to be consistent in the experiment. In the case that the participant uses his one finger to compare two conditions, a sufficiently long interval is needed between the perception tests. This interval may affect the answers, including reducing the accuracy. Therefore, we decided to request the participants to use both hands to detect the differences of thermal perception between the two conditions. The experimental conditions were as follows: contact time to the device—5 s; room temperature—25 °C.

## 4. Results and Discussion

### 4.1. Measurement of the Contact Area

Figure 6 shows the relationship between the amount of injected liquid metal and the contact area between the top surface and the liquid metal via a latex rubber membrane. The horizontal axis shows the injection amount of the liquid metal, and the vertical axis shows the contact area between the liquid metal and the titanium cover. The result shows that the contact area of the liquid metal increased with the injection amount in an almost linear manner. 

### 4.2. Measurement of Thermal Conductivity

Figure 7 shows the relationship between the contact area and the thermal conductivity. The horizontal axis shows the contact area, and the vertical axis shows the measured thermal conductivity. The result indicated that the device could successfully present a wide range of thermal conductivities from 70.3 W/mK to 105 W/mK in a linear manner. 

### 4.3. Evaluation of Thermal Sensation

Figure 8 shows the relationship between the thermal conductivity (horizontal axis) and the average and standard deviation of the scores of the thermal sensation (vertical axis). As the thermal conductivity increased, the scores of the thermal sensation increased, i.e., the participants felt the surfaces cooler. Significant differences were found between the initial state (70.3 W/mK) and 90.7 W/mK. These results verified the effectiveness of the proposed thermal tactile display. Though the case at the thermal conductivity of 90.7 W/mK showed significant difference against the one at 70.3 W/mK, the cases of 93.6 W/mK and 97.1 W/mK did not. All the cases have rather large standard deviation. Ambiguity in the thermal sensation needs to be taken into consideration when designing the thermal sensation display.

The human sensory experiment was conducted at room temperature, and the differences between the body temperature and the room temperature lead to change in the surface temperature of the device. One may consider that the participants detected the thermal sensation based on the surface temperature. However, if this is the case, the device with higher thermal conductivity should represent a warmer sensation, which is contrary to the experimental results. We conclude that in the experiments, the thermal conductivity was the dominant factor in determining the thermal sensation of the participants. 

### 4.4. Perceptual Thershold on Thermal Conductivity

According to Figure 6 and Figure 7, the standard deviation of the contact area and the thermal conductivity is no more than 4.6 mm^2^ and 0.64 W/mK, respectively. From this result, we can say that the thermal conductivity could be controlled precisely by the injection amount of the liquid metal. In the experiments, we prepared two devices for both hands, which had the identical thermal conductivity with the same injection amount of liquid metal. Table 1 and Table 2 show the analysis results of the experiment. The binomial test was used as an analysis method, for which the significance level was 5%. As a result of increasing the thermal conductivity by 5 W/mK (Table 1), when the difference of thermal conductivity between the two devices was 20 W/mK, a difference in the thermal sensation could be perceived. Then, the thermal conductivity was increased by 1 W/mK in the range of 91 W/mK to 94 W/mK. According to Table 2, when the difference of thermal conductivity was 18 W/mK, a difference in thermal sensation could be perceived. Therefore, it can be said that the perception threshold of thermal conductivity is approximately 18 W/mK. However, the perceptual threshold of thermal conductivity might not be the same in other ranges. Therefore, as a future study, we are planning to investigate the perception threshold within different ranges of thermal conductivity by changing the structure of the device. To the best of our knowledge, this is the first time that the thermal perception threshold has been experimentally deduced. 

## 5. Conclusions

We successfully developed a thermal sensation display with controllable thermal conductivity. The encapsulated liquid metal increased the thermal conductivity as it contacts the device’s top surface in a larger area. Control of the effective thermal conductivity with the contact area was successfully demonstrated. Using this device, the threshold of the thermal conductivity necessary to perceive differences in thermal sensation was experimentally deduced to be 18 W/mK. The proposed thermal sensation display can be readily applicable to the presentation of thermal sensations, i.e., hot/warm/cool/cold, for virtual/augmented reality applications and as a research platform for human thermal sensation.

## Figures and Tables

**Figure 1 micromachines-10-00359-f001:**
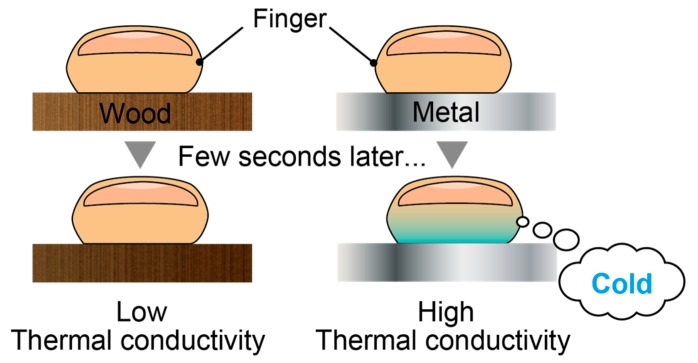
Illustration of heat transfer when we touch wood and metal at the same temperature with fingers.

**Figure 2 micromachines-10-00359-f002:**
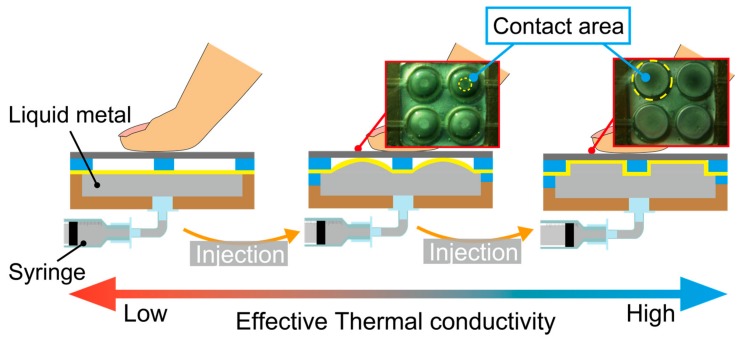
Conceptual sketch of the proposed thermal tactile display.

**Figure 3 micromachines-10-00359-f003:**
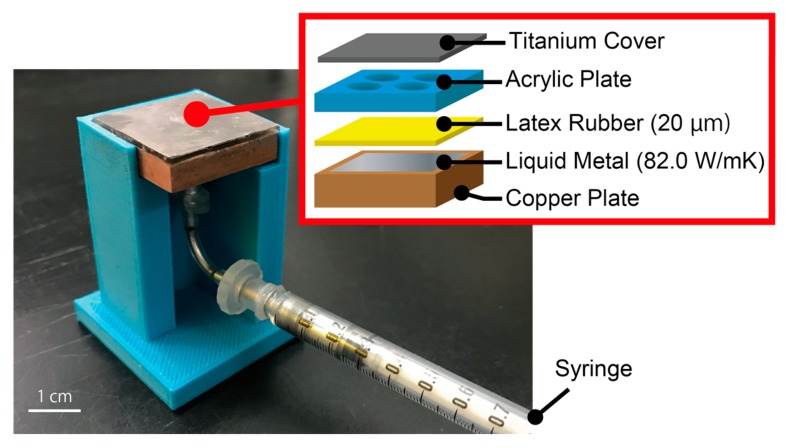
Structure of the thermal sensation tactile display.

**Figure 4 micromachines-10-00359-f004:**
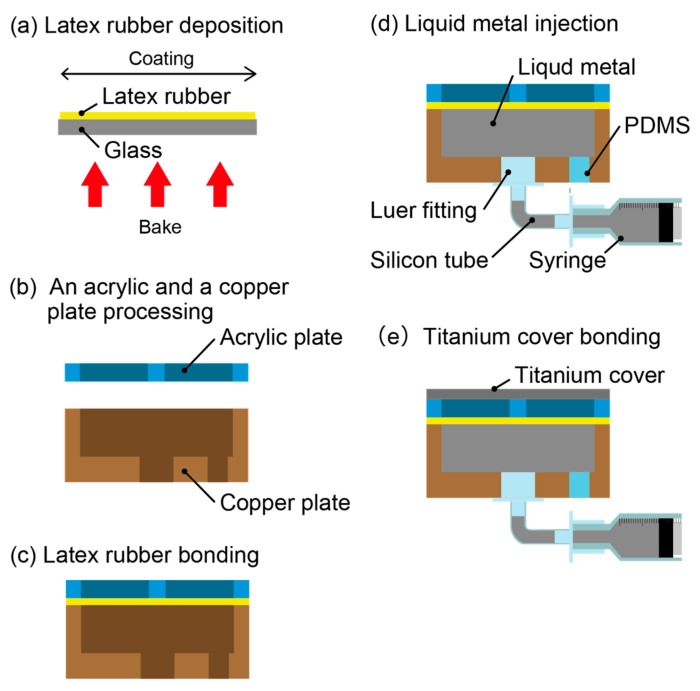
Fabrication process of the thermal sensation tactile display.

**Figure 5 micromachines-10-00359-f005:**
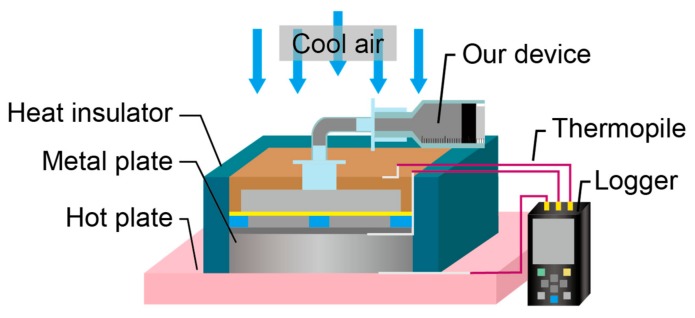
Experimental setup for measurement of thermal conductivity.

**Figure 6 micromachines-10-00359-f006:**
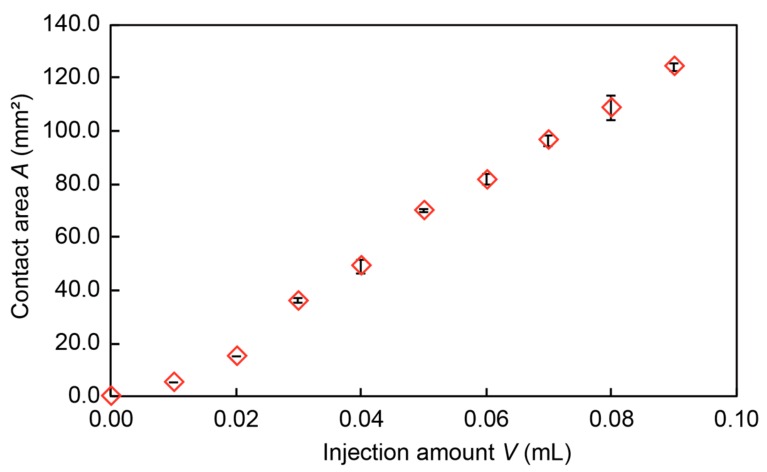
Measurement of the contact area.

**Figure 7 micromachines-10-00359-f007:**
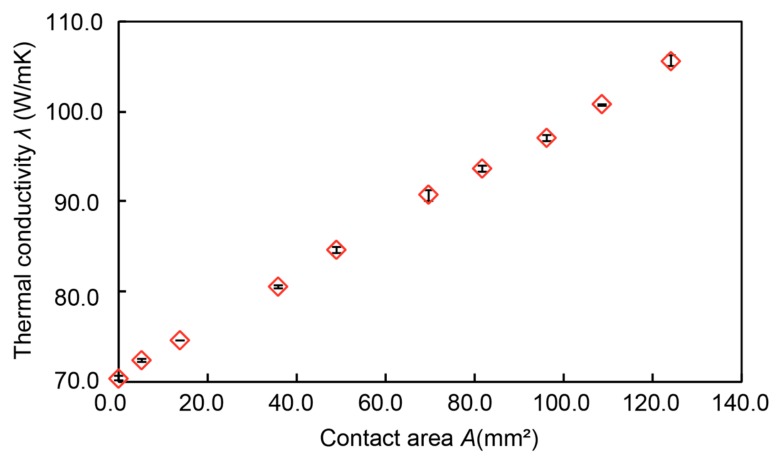
Measurement of thermal conductivity.

**Figure 8 micromachines-10-00359-f008:**
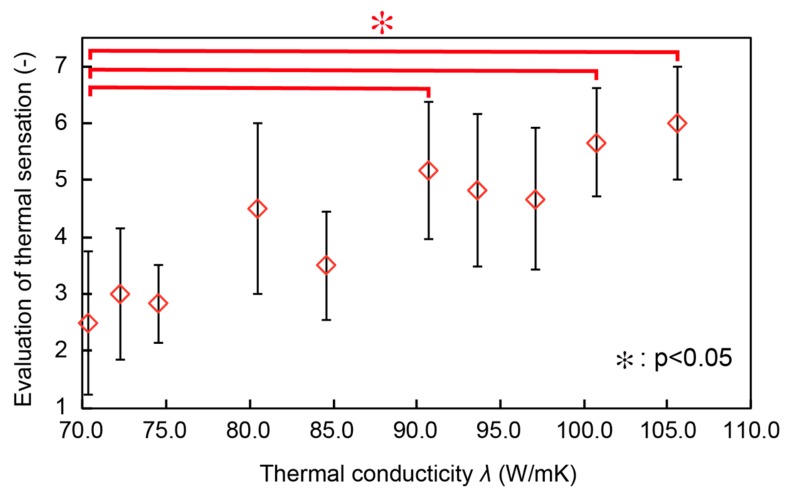
Evaluation of thermal sensation using the semantic differential (SD) method.

**Table 1 micromachines-10-00359-t001:** Result of threshold experiment by using significant different judgement.

Thermal Conductivity (W/mK)	Value Difference (W/mK)	Significant Difference
75 vs. 80	5	NS
75 vs. 85	10	NS
75 vs. 90	15	NS
75 vs. 95	20	*
75 vs. 100	25	*
75 vs. 105	30	***

NS: Not significant; *** *p* < 0.001; * *p* < 0.05.

**Table 2 micromachines-10-00359-t002:** Result of threshold experiment by using significant different judgement.

Thermal Conductivity (W/mK)	Value Difference (W/mK)	Significant Difference
75 vs. 91	16	NS
75 vs. 92	17	NS
75 vs. 93	18	*
75 vs. 94	19	*

NS: Not significant; * *p* < 0.05.

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
