# Peer review of "A Thermal Tactile Sensation Display with Controllable Thermal Conductivity"

_micromachines, 2019, doi:10.3390/mi10060359_

Round 1

Reviewer 1 Report

Dear Authors, 

Thank you for the good presentation of  this work. It seems good to me. However, some notes should be taken into the considerations. Some of them are:

1- The introduction can be improved by adding the last updated of other people work regarding your work and that can be demonstrated by explain the main differences between current work and other people work. 

2- The sentence in line 91 need to be in the same language style. 

3- Since you are working in the micr-scale and the small variations in the temperatures will act, you need to clarify your assumptions. For example, did you consider heat transfer processes by radiation even its effect is very small?

Thank you,

Sincerely, 

Author Response

Thank you for your precious comments. We have prepared the response to the reviewer's comments as a Word file.

Reviewer 2 Report

This paper presents varying the thermal conductivity under the finger-tip as a novel technique for creating thermal sensation. The paper gives a new device with its fabrication and evaluation. It shows controllable thermal conductivity with liquid-metal pumping.

In the perception study, two devices are used. One has fixed thermal conductivity and the other with randomly varying thermal conductivity. The users touch the two devices with their index fingers in both hands. They answer difference in coldness on a seven-point Likert scale.    

While the novelty of the method and the device are clear. The results need further clarification and presentation to convince the readers that the method actually works and that the claimed perception threshold is conclusive.

Figure 1 illustrates the cooler sensation under the finger-tip due to higher thermal conductivity. However, the experiment in Figure 5 shows the changes in thermal conductivity. It does not show the change in temperature at the touch surface due to change in thermal conductivity. A further experiment might be required without the hot plate and cool air to show that the temperature at the touch surface changes as a result of change in thermal conductivity. This could complete the prototype evaluation. Note that in the user-study, there is no hot plate and cool air used. Currently, the change in surface temperature is left to the users to sense in the user-study. This could be acceptable but could also be clarified and discussed.

Could a citation be provided for the two-point identification method used in the paper? Detecting thermal sensation difference using both hands simultaneously but separately seems tricky. Could you refer to studies where both hands were used to differentiate tactile sensation? The study gives one hand a random cooler (not hotter) sensation than the other hand which has the fixed but minimum thermal conductivity. Please provide a reference to argue that there is no bias in the study due to this.

It is not clear in the text that the condition of equal thermal conductivity on both the devices were used in the study as shown in Figure 8.

In Figure 8, the thermal conductivity in the x-axis will make more sense as it is said in the text that the thermal conductivity is varied in the experiment by 5 W/mK.

However, it seems that the liquid-metal injection volume is varied by 0.01 ml. This is confusing as varying the thermal conductivity is also mentioned. The relationship is not exactly linear as seen in the Figures 6 and 7.  

There is inconsistency between Figure 8 and Table 1 because the injection volume is shown in the x-axis of Figure 8, but thermal conductivity is shown in the first row of Table 1.

Table 1 does not show the result of threshold experiment for all the data points. It only shows up to 0.05 ml. However, the thermal sensation data for 0.06 and 0.07 ml are also close to the data for 0.03 ml which is not significant (NS). With the partial presentation of data, the perception threshold conclusion is not convincing. This is my ***main concern***. Please show the significance difference for all the data points.      

Another figure similar to Figure 8 is required to show the evaluation of thermal sensation data where the thermal conductivity was varied with a step of 1 W/mK.

Will the perception threshold vary if the study is conducted around another mean thermal conductivity, say 150 W/mK instead of 75 W/mK?

The paper would be interesting to the users if a practical implementation on a usable device could be discussed.

Author Response

(The authors gave the same response as above.)
